# Potential Diagnostic and Clinical Significance of Selected Genetic Alterations in Glioblastoma

**DOI:** 10.3390/ijms25084438

**Published:** 2024-04-18

**Authors:** Silvia Tomoszková, Jozef Škarda, Radim Lipina

**Affiliations:** 1Neurosurgery Clinic, University Hospital Ostrava, 17. listopadu 1790/5, 708 00 Ostrava, Czech Republic; radim.lipina@fno.cz; 2Medical Faculty, University of Ostrava, Syllabova 19, 703 00 Ostrava, Czech Republic; jozef.skarda@fno.cz; 3Institute of Molecular and Clinical Pathology and Medical Genetics, University Hospital Ostrava, 17. listopadu 1790/5, 708 00 Ostrava, Czech Republic

**Keywords:** glioblastoma, prognosis, gene mutations, overall survival

## Abstract

Glioblastoma is currently considered the most common and, unfortunately, also the most aggressive primary brain tumor, with the highest morbidity and mortality rates. The average survival of patients diagnosed with glioblastoma is 14 months, and only 2% of patients survive 3 years after surgery. Based on our clinical experience and knowledge from extensive clinical studies, survival is mainly related to the molecular biological properties of glioblastoma, which are of interest to the general medical community. Our study examined a total of 71 retrospective studies published from 2016 through 2022 and available on PubMed that deal with mutations of selected genes in the pathophysiology of GBM. In conclusion, we can find other mutations within a given gene group that have different effects on the prognosis and quality of survival of a patient with glioblastoma. These mutations, together with the associated mutations of other genes, as well as intratumoral heterogeneity itself, offer enormous potential for further clinical research and possible application in therapeutic practice.

## 1. Introduction

Glioblastoma (GBM) is currently considered the most common and unfortunately also the most aggressive primary brain tumor, with the highest morbidity and mortality rate. The average survival of patients diagnosed with GBM is 14 months [1], and only 2% of patients survive 3 years after surgery [2]. A basic problem in treating GBM is the lack of adequate prognostic and predictive markers that could help to reveal mechanisms of tumorigenic transformation. The present literature describes only some of the changes associated with a better response of GBM patients to therapy. However, it should be noted that, so far, most described prognostic factors have been negative ones (deletion 9p, loss of chromosome 10, amplification of the *EGFR* gene).

Current treatment protocols for GBM are ineffective due to the invasive growth of the tumor, which does not allow for complete surgical resection. Another major problem in the treatment of GBM is its resistance to radio- and chemotherapy.

According to extensive clinical studies, patient survival is mainly related to the molecular biological properties of glioblastoma, which are of interest to the general medical community.

Many studies have been published in connection with the molecular genetic mechanisms of GBM transformation, based mainly on retrospective analyses of GBM samples. However, no panel of genes has yet been found to allow an effective therapeutic stratification of GBM patients.

Recent work has identified 20 genes that appear to play a role in the development of diffuse gliomas. Mutations of these genes can trigger molecular changes that contribute to tumor formation. These genes include: *ATRX*, *BRAF*, *CDKN2A*, *CDKN2B*, *CDKN2C*, *CIC*, *EGFR*, *FUBP1*, *H3F3A*, *IDH1*, *IDH2*, *NF1*, *NF2*, *NRAS*, *PIK3CA*, *PIK3R1*, *PTEN*, *RB1*, *TERT*, and *TP53* [3]. Understanding the specific roles of these genes and the impact of their mutations is crucial for developing new treatment strategies for glioblastoma patients.

In this article, we have explored the role of 11 selected genes in the pathophysiology of glioblastoma because they are the subject of interest in several studies, and some of them are not routinely investigated. In addition, these genes are interconnected at the level of signaling pathways.

There are three most important signaling cascades frequently deregulated in glioblastoma: PIK-AKT-mTOR, p14ARF-MDM2-MDM4-p53, and CDK4/6-CDKN2A/B-RB1 cell-cycle pathway [4].

The PIK pathway is important for regulating cell growth. Signaling is activated by RTKs and/or RAS and inhibited by *PTEN* [4]. *PTEN* mutations are believed to exert their effects through the putative PI3K-AKT-mTOR signaling pathway. Specifically, loss of *PTEN* leads to activation of AKT, which in turn promotes anti-apoptotic and pro-cell cycle entry pathways believed to be essential in tumorigenesis [5].

*EGFR* induces mTORC2 kinase activity, which is partially restricted by phosphatase and tensin homolog *PTEN*. The mTORC2 signaling enhances GBM growth and survival [6]. *EGFR* aberrations over-activate downstream pro-oncogenic signaling pathways, including the RAS-RAF-MEK-ERK MAPK and AKT-PI3K-mTOR pathways [7].

The p53 pathway is central to the induction of DNA repair, cell-cycle arrest, and apoptosis. CDK4 and CDK6 catalyze the phosphorylation of RB1 and, thereby, the release of E2 transcription factors, which then induce the expression of genes involved in the progression from the G1 to S phase of the cell cycle [4].

*PIK3CA* encodes the p110α catalytic subunit of PI3K, a key enzyme in the PI3K/AKT/mTOR signaling pathway. Mutations in *PIK3CA* lead to increased PI3K activity, resulting in downstream activation of AKT and mTOR [8].

## 2. Methods

Our study examined a total of 71 retrospective studies published from 2016 through 2022 and available on PubMed that deal with mutations of selected genes in the pathophysiology of glioblastoma. We used the search terms concrete gene mutation, prognosis, and quality of survival of patients with GBM and then summarized the results. We discuss the 11 most common gene alterations and their potential diagnostic and, especially, clinical significance. We selected these 11 genes for two key reasons. First, they are the subject of frequent studies, indicating their potential importance. Second, these genes are interconnected within a network of signaling pathways, suggesting they may cooperate or influence each other in complex ways. The articles involved in this review were carefully selected to meet the criterion of relevancy and were studies performed in larger clinical centers on a larger cohort of glioblastoma patients (from 5 to 1308, with a median of 149.5 patients). While case reports were generally excluded, we included an exception for the work of Takahashi et al. [9]. This report described a young GBM patient with a *BRAF* mutation who exhibited a prolonged survival of 48 months. We averaged the resulting values of overall survival in months and divided patients into 4 groups according to survival time: group A, with an average survival time of over 25 months; group B, 20–25 months; group C, 15–20 months; and group D, under 15 months. The remaining references address the molecular biology of selected genes.

## 3. Results

### 3.1. ATRX

The ATRX transcriptional regulator, also known as ATP-dependent ATRX helicase, X-linked helicase II, or X-linked nuclear protein (XNP), is encoded by the *ATRX* gene in humans [10].

*ATRX* is located on chromosome Xq21.1 and encodes a 280 kDa nucleoprotein that is involved in many cellular functions, including DNA recombination and repair and chromatin remodeling [11]. ATRX undergoes cell cycle-dependent phosphorylation, which regulates its nuclear matrix and involvement in regulatory mechanisms in the cell [12].

Hereditary mutations of *ATRX* have been described in association with XLMR syndrome (X-linked mental retardation syndrome) and alpha-thalassemia. Acquired mutations have been reported in various types of human cancer, such as pancreatic neuroendocrine tumors, gliomas, astrocytomas, and malignant pheochromocytomas [13]. In many studies, *ATRX* gene alterations have been shown to be associated with prognosis and mostly correlate with favorable results [14].

Studies by Pekmezci et al. [14] and Han et al. [15] suggest that specific characteristics of *ATRX* loss of expression are linked to the response to temozolomide treatment. Glioblastoma patients lacking *ATRX* expression appear to benefit more from this treatment [15].

Mutations in the *IDH*, *ATRX*, and *TERT* promoter and the correlations between them were analyzed in a study conducted by Ohba et al. [16]. Immortalized cells overcome the telomere-related crisis by activating telomerase or the ALT process (alternative lengthening of telomeres) [17]. In gliomas, telomerase is activated mainly by a mutation in the *TERT* promoter, while ALT activation is usually associated with an *ATRX* mutation. Although the mechanism used by the *ATRX* mutation to induce ALT remains unclear, the loss of *ATRX* alone is considered insufficient to induce ALT [16,17].

Tumor cell lines using ALT to maintain the length of the telomere usually show complex karyotype rearrangements consistent with genome instability that can occur with dysfunctional telomeres. The absence of the ATRX chromatin remodeling factor is the dominant prognostic marker in these types of cancer [17].

A study by Liu et al. [18] examined the association between *TERT* and *ATRX* mutations. Their findings provide a theoretical basis for further research and may improve the clinical diagnosis and treatment of gliomas in the future. They showed that mutations in the *TERT* promoter are negatively associated with *ATRX* expression in GBM. They also studied the effects of *IDH* and *TERT* mutations, Ki-67 protein expression, and age on *ATRX* status. Sex, WHO grade, and Ki-67 expression did not appear to significantly affect *ATRX*. However, age and *IDH* mutations were found to be statistically related. The probability of *ATRX* mutation increased by 8.8% for each additional year of age. Moreover, the probability of *ATRX* mutation in *IDH*-mutant GBM samples was 14 times higher than in *IDH*-wild-type samples [19]. They furthermore confirmed that *TERT* promoter mutations are positively associated with age and WHO grade, but they worsen the overall prognosis in association with the *ATRX* mutation present [18,19].

Bobeff et al. [20] determined the levels of amino acids in the plasma of 18 patients diagnosed with GBM and in a control group of 15 healthy volunteers by liquid chromatography and LC-QTOF-MS spectroscopy. Phenylalanine and leucine levels were shown to be lower in patients with GBM if *ATRX* gene expression was lost. They thus showed that the levels of free amino acids in the plasma of patients with GBM differ significantly from the levels in healthy people, so they can be used as a prognostic marker [20].

Gulten et al. [21] also addressed the loss of *ATRX* expression. They performed a retrospective analysis of 83 patients with GBM to determine *ATRX* and *IDH1* mutations and p53 expression. Of the entire group, *IDH1* mutation was detected in 9.6% of patients, *ATRX* loss in 4.8%, and p53 expression in 12.05% [21]. It was found that *IDH1* mutation, loss of *ATRX*, and *p53* expression alone did not have a major impact on patient prognosis, but radiotherapy and chemotherapy have a positive effect on the survival of patients with these mutations [21].

Several studies have identified positive correlations between patient survival and the presence of specific genetic alterations in glioblastoma. These alterations include mutations in the *IDH1* gene, expression of the *TP53* protein, and loss of expression of the *ATRX* gene [22,23,24].

Chaurasia et al. [22] further categorized glioblastoma patients into three prognostic subgroups based on these markers: group 1: lacking both *IDH1* mutation and *TP53* expression; group 2: possessing *IDH1* mutation and *ATRX* loss; group 3: having IDH1 mutation but lacking TP53 expression. Their findings suggest that patients in groups 2 and 3, characterized by specific combinations of these markers, exhibit improved survival outcomes [22].

Separate research by Cai et al. [23] investigated the expression of mutated *ATRX* and *IDH1* alongside heat shock proteins (hsp27 and P-hsp27) in a large sample of GBM patients. While they observed elevated levels of hsp27 and P-hsp27 in aggressive types of glioblastoma, these proteins did not significantly impact patient survival [16]. This finding led to the classification of glioblastoma into three distinct groups: group A: GBM with both *IDH1* mutation and *ATRX* loss, group B: GBM expressing P-hsp27, and group C: all remaining samples.

Analysis of survival data revealed the longest average survival (19.6 months) in group A, followed by group B (15 months) and group C (13 months). These results support the notion that GBMs harboring both *IDH1* mutation and *ATRX* loss exhibit a more favorable prognosis. Interestingly, the presence of P-hsp27 within group A (*IDH1* mutation and *ATRX* loss) further improved patient outcomes [25].

ATRX is involved in the replication and repair of damaged DNA. The CRISPR/Cas9-mediated genetic activation of *ATRX* inhibits cell proliferation and angiogenesis, and genetic activation of *ATRX* may serve as a prognostic marker in predicting sensitivity to temozolomide [14,15].

### 3.2. BRAF

*BRAF* is a human gene located on the long arm of chromosome 7 (7q34), encoding a serine-threonine protein kinase called B-Raf [26]. This protein plays a role in regulating the MAP kinase/ERK signaling pathway, which affects cell division, differentiation, and secretion.

Increased BRAF kinase activity causes the sustained activation of the MAPK signaling pathway with consequent increased rates of proliferation and long-term survival of tumor cells. The basis of successfully targeted treatment is, therefore, the inhibition of BRAF kinase to slow or stop the growth of tumor cells. This is possible thanks to BRAF inhibitors, including the protein kinase inhibitor vemurafenib (Zelboraf), which is used in the treatment of malignant melanoma [27]. Research by Kleinschmidt-DeMasters et al. [28] indicates the potential effectiveness of vemurafenib in treating *BRAF*-mutant GBM [28].

*BRAF* gene mutations can cause birth defects in cardiofaciocutaneous syndrome, which is characterized by heart defects, mental retardation, and facial changes. Acquired *BRAF* mutations have been found in non-Hodgkin’s lymphoma, colorectal cancer, malignant melanoma, and brain tumors, including glioblastoma [29].

Several studies have investigated the link between *BRAF* mutations and prognosis in glioblastoma patients, particularly in younger ones [30,31,32]. Zheng et al. [30] found a higher prevalence of BRAF mutations in young women with glioblastoma. Their study included 16 patients aged 7–61, with all 16 exhibiting *BRAF* mutations. Notably, younger patients in this study also showed a better prognosis [30].

This age-dependent effect was further supported by Vuong et al. [31]. Their research demonstrated a more favorable prognosis for children and young adults (under 35 years old) with GBM and proven *BRAF* mutation. However, the presence of a BRAF mutation did not significantly impact prognosis in older patients [31].

Takahaschi et al. [9] also showed that *BRAF* mutations are less common in adult patients with GBM. In their study, *BRAF* mutation occurred in only one case, and this patient survived for 4 years after surgery [9]. Similar results were obtained by Chi et al. [32], where the survival of patients with proven *BRAF* mutation was 16–36 months [32].

In a study by Da et al. [33], the activation of the RAS/RAF signaling pathway was found to play a critical role in the pathophysiology of GBM. They reported that it does not matter whether *BRAF* mutation (m-*BRAF*) or *BRAF* amplification (a-*BRAF*) is considered: both activate the RAS/RAF pathway but have different effects on the survival rate of patients with GBM. The a-*BRAF* group had poorer survival than did the m-*BRAF* group [33].

These findings suggest a potential association between *BRAF* mutations, younger age, and improved prognosis in GBM patients. However, further research is necessary to fully understand this correlation and its underlying mechanisms. While some studies have reported a positive correlation between *BRAF* mutations and patient survival [9,30,31,32], others have observed contrasting results. For instance, Wang et al. [34] found *BRAF* mutations in a high percentage (80%) of their GBM samples and linked them to a significant decrease in survival [34].

### 3.3. EGFR

The epidermal growth factor receptor (EGFR) is a transmembrane protein encoded by a gene located on the short arm of chromosome 7 at 7p11.2 [35]. EGFR belongs to the ErbB receptor group, which consists of four closely related receptor tyrosine kinases: EGFR (ErbB-1), HER2/neu (ErbB-2), HER3 (ErbB-3), and HER4 (ErbB-4).

Under physiological circumstances, EGFR is essential for the ductal development of the mammary gland. EGFR agonists such as amphiregulin, TGF-α, and heregulin induce ductal and lobuloalveolar development, even in the absence of estrogen and progesterone [36].

*EGFR* mutations affecting the expression of this gene can lead to uncontrolled cell growth and cancer [37]. *EGFR* mutations leading to its overexpression or amplification are associated with a number of cancers and are often observed in GBM [38]. However, targeted therapy against EGFR has not yet shown any clear clinical benefit. Several factors, including limited blood–brain barrier penetration, tumor cell diversity within the mass (intratumoral heterogeneity), and activation of alternative signaling pathways, can all contribute to the tumor’s resistance to this treatment [39].

*EGFR* amplifications, commonly observed in GBM, have been shown by Matini et al. to promote angiogenesis and vascular proliferation within the tumor [40] and are associated with poor survival [40,41]. Drugs that could block the EGFR pathway could, therefore, be useful in the treatment of GBM [40].

While research by Matini et al. [40] highlights the potential role of *EGFR* amplification in GBM progression, other factors also influence patient outcomes. Armocida et al. [42] conducted a retrospective study to evaluate the prognostic impact of *EGFR* amplification in wild-type GBM samples of children and adults [42]. They compared the amplification of *EGFR* with various clinical factors, including patient age, tumor volume, and overall survival. Interestingly, their findings revealed a strong correlation between patient age and tumor volume with overall survival, suggesting these factors may be more significant prognostic indicators [42].

Our analysis, along with findings from studies by Munoz-Hidalgo et al. [43] and Schaff et al. [44], consistently points towards a negative impact of *EGFR* alterations on glioblastoma patient outcomes. Munoz-Hidalgo et al. [43] observed poorer survival in patients with confirmed *EGFR* amplifications [43], and Schaff et al. [44] identified a significant correlation between *EGFR* amplification and *MGMT* methylation status, which can influence treatment response [44].

A study by Weller et al. [45] investigated whether adding the vaccine rindopepimut to standard temozolomide chemotherapy could improve patient outcomes in newly diagnosed GBM. Unfortunately, the study found no significant impact of rindopepimut on overall survival in these patients [45].

Targeted immunotherapy holds promise as a future treatment strategy for GBM, but further research is needed to realize its full potential.

### 3.4. IDH1, IDH2

Isocitrate dehydrogenase is an enzyme that catalyzes the reversible oxidative decarboxylation of isocitrate to α-ketoglutarate (α-KG) as part of the tricarboxylic acid cycle in glucose metabolism in cells. This step also allows for the simultaneous reduction of nicotinamide adenine dinucleotide phosphate (NADP^+^) to reduced nicotinamide adenine dinucleotide phosphate (NADPH) [46]. It is, therefore, involved in energy metabolism.

*IDH1* is located on the long arm of chromosome 2 at 2q34, while *IDH2* is on the long arm of chromosome 15 at 15q26.1.

Because NADPH and α-KG have detoxification functions in the cell in response to oxidative stress, IDH1 is also indirectly involved in alleviating oxidative damage. In addition, IDH1 is key to the β-oxidation of unsaturated fatty acids in liver cell peroxisomes and is also involved in the regulation of glucose-induced insulin secretion [47,48]. Notably, IDH1 is the primary producer of NADPH in most tissues, especially in the brain. IDH2 has similar functions.

Mutations in the *IDH* gene, specifically those affecting arginine residue R132, which are the most common, lead to a reduced ability to convert isocitrate to ketoglutarate, thereby reducing levels of ketoglutarate and NADPH, making a cell more sensitive to oxidative stress [49]. Alteration of the enzyme’s binding site results in loss of normal enzymatic function and abnormal production of 2-hydroxyglutarate (2-HG), which inhibits the enzymatic functions of many alpha-ketoglutarate-dependent dioxygenases. These include histone and DNA demethylases. Consequently, their inhibition results in extensive changes in histone methylation and DNA itself and thus promotes tumorigenesis [50,51].

Mutations in the *IDH1* and *IDH2* genes strongly correlate with the development of glioma, acute myeloid leukemia, chondrosarcoma, intrahepatic cholangiocarcinoma, and angioimmunoblastic carcinomas of T-cell lymphomas [52]. Tumors of various types with *IDH1/2* mutations show better responses to radiotherapy and chemotherapy [53,54].

The 2021 revision of the World Health Organization (WHO) classification system for central nervous system tumors reclassified IDH-mutant glioblastoma as IDH-mutant astrocytoma, WHO grade 4. This distinction reflects the improved prognosis associated with IDH mutations, as these tumors typically exhibit a better response to radiotherapy and chemotherapy compared to their IDH-wild-type counterparts.

*IDH1* mutation is found in 5.6–12% of patients with gr. 4 astrocytoma. Chen et al. [55] evaluated *IDH1* mutations in a sample of 1011 astrocytomas (previously glioblastomas). *IDH1* mutation was detected in 570 patients. Patients with proven *IDH1* mutation were found to have a better prognosis, with a median survival of 1.1–3.7 years [55].

Emerging evidence from various research groups suggests a positive association between *IDH* mutation status and improved clinical outcomes, particularly in terms of extended overall survival rates, for grade 4 astrocytoma patients [54,56,57,58].

### 3.5. MGMT

The *MGMT* gene is located on the long arm of chromosome 10 at 10q26.3. This gene encodes O-6-methylguanine-DNA-methyltransferase, an enzyme important for genome stability. It repairs damaged guanine nucleotides by transferring the methyl group at the O6 guanine site to its cysteine residues, thus preventing gene mutation, cell death, and tumorigenesis caused by alkylating agents. It removes alkaline groups, which are an important part of guanine methylation, from guanine. Therapy with the alkylating cytostatic drug temozolomide is based on this principle, assuming that the methylated form is nonfunctional.

*MGMT* methylation is observed in patients with glioblastoma [59] but more often in anaplastic oligodendrogliomas [60]. Studies have shown a positive effect of proven methylation of the *MGMT* promoter in patients with high-grade glioma on the overall prognosis [61,62,63,64,65,66,67,68].

*MGMT* promoter methylation is, therefore, an important genetic alteration that has received significant research attention. It is associated with a more favorable response to temozolomide, a standard chemotherapy drug used for GBM [69,70].

Li et al. [71] investigated a group of GBM patients who received temozolomide therapy and categorized them based on treatment response (progression, non-progression, pseudo-progression). Notably, *MGMT* promoter methylations were more prevalent in the pseudo-progression group, where initial scans suggest tumor growth but may not reflect true tumor recurrence. This finding suggests that *MGMT* methylations might be associated with a pseudo-progression phenomenon, requiring careful monitoring to avoid unnecessary treatment changes. Interestingly, the pseudo-progression group also exhibited a longer average survival time compared to the early-progression group [71].

Correlations between *MGMT* promoter methylation and *TERT* mutations were the focus of other studies [72,73,74]. Arita et al. [72] showed worse outcomes in *TERT*-mutant GBMs lacking *MGMT* methylation. Similarly, Vuong et al. [73] reported a survival benefit associated with *TERT* mutations only in *MGMT*-methylated tumors via meta-analysis. Shu et al. [74] identified *MGMT* methylation and *TERT* mutations as independent prognostic factors, with these alterations, along with clinical features, forming distinct prognostic subgroups.

### 3.6. PIK3CA

*PIK3CA* is a gene located on the long arm of chromosome 3 at 3q26.32 and encodes phosphatidylinositol-3-kinase (also referred to as p110α). Phosphatidylinositol-3-kinases belong to the group of lipid kinases and are responsible for phosphorylating the 3-OH residue of the inositol ring of phosphoinositides, thereby being involved in the coordination of various cellular functions, including proliferation.

The *PIK3CA* gene has been shown to be oncogenic and involved in the pathophysiology of cervical, breast, and colorectal cancer [75]. Gallia et al. [76] have shown that *PIK3CA* mutations occur in a large number of glioblastoma patients and are, therefore, of therapeutic importance [76].

McNeill et al. [77] demonstrated that *PIK3CA* mutations are limited to three functional domains: the adapter binding domain and the helical and kinase domains. Defining how these mutations affect gliomagenesis and the response to kinase inhibitor therapy (PIK3i, MEKi) may help in the development of new targeted therapies in patients with GBM [77].

Tanaka et al. [78] observed activating mutations in the *PIK3CA* gene in 6–15% of glioblastomas. They retrospectively analyzed a group of 91 patients with GBM, with a mean age of 58 years (23–85), median PFS of 11.9 months, and median overall survival of 24 months [78]. Thirteen patients (8.3%) had a proven *PIK3CA* mutation. *PIK3CA* mutation was associated with younger age (mean 49.4 years) and correlated with shorter PFS (6.9 months) and shorter overall survival (21.2 months). An association between *PIK3CA* mutation and multiple disseminated disease, multiple lesions, or leptomeningeal spreading was observed in 46.2% of patients [78].

### 3.7. PIK3R1

*PIK3R1* is a gene located on the long arm of chromosome 5 at 5q13.1. It has similar functions as *PIK3CA*. It encodes a phosphatidylinositol-3-kinase and plays an important role in the metabolic action of insulin. Mutations in the gene are associated with insulin resistance.

Mutations in the *PIK3R1* gene have been addressed by Quayle et al. [79]. The authors found that mutations in the iSH2 pathway of PIK3R1 trigger oncogenic activity, and thus, patients with proven *PIK3R1* mutation can benefit from treatment with AKT inhibitors [79]. Somatic mutations in *PIK3R1* are observed in many types of tumors.

The tumorigenic activity of *PIK3R1* has been demonstrated in GBM. Weber et al. [80] mapped changes in the *PIK3CA* and *PIK3R1* genes. They found that eliminating either of these genes alone in GBM cell lines by lentivirus-mediated shRNA expression resulted in reduced proliferation, migration, and invasion in all the cells tested [80].

Mutations in *PIK3CA* and *PIK3R1*, key components of the PI3K signaling pathway, are emerging as potential therapeutic targets in GBM due to their critical role in regulating cell growth and activity [80].

### 3.8. PTEN

The *PTEN* (phosphatase and tensin homolog) gene is located on the long arm of chromosome 10 at 10q23.3. It encodes a protein with phosphatidylinositol-3,4,5-trisphosphate 3-phosphatase activity, the activity of which attenuates AKT/PKB cascade signaling. It is thus a tumor suppressor gene and is currently the target of many anti-cancer drugs [81]. *PTEN* gene mutations are especially involved in the pathophysiology of glioblastomas and lung, breast, and prostate tumors [81].

Koshiyama et al. [82] evaluated *PTEN* and *TP53* mutations in a cohort of 40 glioblastoma patients with a median age of 59.3 years (range 41–83 years) and a male predominance (70%). The median survival was 145 days. *EGFR* amplification was detected in 42.5%, *PTEN* deletion in 35%, and *TP53* deletion in 22.5% of patients. Notably, confirmed *TP53* and *PTEN* mutations were associated with a poorer prognosis [82].

Xu et al. [83] investigated the influence of *PTEN* mutations on progression-free survival (PFS) in a larger cohort of 586 GBM patients. *PTEN* mutation status is recognized as a factor affecting treatment response and relapse risk. Among *PTEN* mutations, the authors describe missense, nonsense, frameshift, and other types of mutations [83]. Their frequencies and associated PFS are detailed in Table 1.

Analysis of *PTEN* mutations in glioblastoma patients reveals a potential link between mutation type and prognosis, as reported by Xu et al. [83]. Their study demonstrates that nonsense mutations have a more significant negative impact on progression-free survival compared to other mutation types (potentially associated with *PTEN* overexpression). The median PFS for patients with nonsense mutations was 3.8 months, whereas patients with other mutation types exhibited a median PFS of 7.2 months. Interestingly, missense/frameshift mutations did not appear to have a substantial influence on PFS. These findings suggest that the specific type of *PTEN* mutation may influence patient outcomes [83].

Overall, the presence of a *PTEN* mutation, regardless of type, has been associated with a decrease in PFS by up to 50% [83]. This highlights the potential prognostic value of *PTEN* mutation analysis in GBM patients.

Carico et al. [84] investigated the potential association between *PTEN* mutation status and overall survival in GBM patients. Their study included 155 patients, with 65% harboring confirmed *PTEN* mutations. The *PTEN*-mutant group had a mean age of 63 years, and 70% of patients scored above 80 on the Karnofsky Performance Scale (KPS), indicating a good performance status. Although no significant associations were found between individual patient criteria and *PTEN* mutation or its overexpression alone, patients with *PTEN* deletion were generally older, had a higher degree of neurological impairment, and were undergoing a less extensive surgical resection. The authors further analyzed the impact of various characteristics potentially linked to *PTEN* status on overall survival. Details regarding these characteristics and their association with OS are presented in Table 2 of the original study [84].

Younger patients (under 65 years old) with a higher KPS score (over 80) and a large extent of resection (total or gross total resection) are important predictors of further patient survival [84].

The prognostic significance of *PTEN* mutations in glioblastoma remains a topic of ongoing investigation, with conflicting results reported in the literature. Several studies, including those by Kraus et al. [85], Tadipatri et al. [86], and Ermoian et al. [87], have identified *PTEN* mutations as a negative prognostic factor, associating them with shorter survival times [85,86,87]. For instance, Ermoian et al. reported a median survival of 195 weeks (10–411) in glioblastoma patients with *PTEN* mutation [87].

However, other studies have challenged this association. Ruano et al. [88] did not observe a significant impact of *PTEN* mutation on prognosis, suggesting that mutations in *EGFR* and *TP53* might be more relevant for predicting survival. Their study reported a median overall survival of 10 months for the entire glioblastoma patient cohort [88]. Backlund et al. [89] also reported a median survival of 437 days in their group of GBM patients, with *PTEN*-mutant glioblastoma having a median survival of 166 days [89].

These contrasting findings highlight the complexity of *PTEN* mutations and their potential interaction with other genetic alterations in influencing GBM prognosis. Further research is necessary to elucidate the precise role of *PTEN* mutations and identify robust prognostic markers for GBM patients.

### 3.9. TERT

Telomerase reverse transcriptase (TERT) is a catalytic subunit of the telomerase enzyme. The *TERT* gene is on chromosome 5.

Reverse transcriptases are enzymes that catalyze the process of transcribing genetic information from ribonucleic acid (RNA) to deoxyribonucleic acid (DNA). In practice, it is most often part of the transmission of the genetic code of a retrovirus such as HIV to the host infected cells’ DNA. Telomerase lengthens telomeres in DNA strands, allowing aging cells that would otherwise undergo apoptosis to exceed the Hayflick limit and become potentially immortal, as we often see in tumor cells. Almost all GBM show telomerase activity, which is a major agent in achieving cell immortalization [90].

If a correlation between increased telomerase activity and malignancy is demonstrated, the inhibition of the enzyme could induce cell aging and, thus, apoptosis, which could be used in therapeutic practice. Confirmed *TERT* mutation correlates with poorer survival rates.

Nonoguchi et al. [91] investigated the co-occurrence of genetic alterations in GBM, finding a correlation between *TERT* promoter mutations and mutations in *IDH1*, *TP53*, and *EGFR* amplification. Interestingly, their analysis of 358 GBM samples revealed a mutually exclusive relationship between *TERT* and *IDH* mutations, with co-occurrence detected in only a small percentage (3%) of patients [91].

*TERT* expression can be altered by activating mutations in the rs2853669 polymorphism in the promoter region. Spiegl-Kreinecker et al. [92] investigated the prevalence of *TERT* promoter mutations in a cohort of 126 GBM samples. A high frequency (73%) of *TERT* promoter mutations was identified. Among these mutations, C228T and C250T were the most common, detected in 66 and 26 patients, respectively. Details regarding the distribution of other mutation types are presented in Table 3 of the original study [92].

Spiegl-Kreinecker et al. [92] also investigated the rs2853669 single nucleotide polymorphism (SNP) within the *TERT* promoter region. Analysis of their GBM samples revealed that 59 patients (45%) did not harbor this polymorphism, while 67 (53%) did. Interestingly, among the 67 patients with the polymorphism, 12 possessed the homozygous CC genotype, and 55 exhibited the heterozygous CT genotype. Further details regarding the distribution of other genotypes, if any, are found in Table 4 of the original study [92].

The authors did not find a significant correlation between the mutation status of the *TERT* promoter and rs2853669 polymorphism. Proven *TERT* mutation negatively affected prognosis and shortened survival time, especially in the group of patients over 65 years of age. In agreement with Nonoguchi et al. [91], they found *TERT* and *IDH1* mutations to be mutually exclusive [91].

Mosrati et al. [93] evaluated *TERT* promoter mutations and rs2853669 polymorphism in GBMs. The mutation C228T was confirmed in 75% and C250T in 25% of patients. The overall survival time of a patient with proven *TERT* promoter mutation was 11–20 months [93].

Similarly, other authors, including Simon et al. [94], describe *TERT* mutation as a negative prognostic marker [94]. In their group of 147 patients with IDH-wild-type GBM, Kikuchi et al. [95] confirmed *TERT* mutation in 92 patients (62.6%). The median age at diagnosis was 66 years, and patients with *TERT* mutation had a shorter PFS (7–10 months) [95].

Promoter mutations (particularly in the *TERT* promoter) are associated with increased telomerase activity. Fan et al. [96] demonstrated that there are alternative pathways to telomere extension in GBM and that these correlate with *ATRX* mutations. Mutations in *TERT* (telomerase reverse transcriptase promoter) and *ATRX* may allow tumor cells to escape apoptosis [96].

### 3.10. TP53

The *TP53* gene is located on human chromosome 17p13.1. It encodes the p53 protein that consists of 393 amino acids [97,98]. The tumor suppressor and transcription factor p53 plays critical roles in tumor prevention by orchestrating a wide variety of cellular responses, including damaged cell apoptosis, maintenance of genomic stability, inhibition of angiogenesis, and regulation of cell metabolism and tumor microenvironment [98]. The importance of the *Tp53* gene as a tumor suppressor is highlighted in human cancer, where it is the most commonly mutated gene [99]. The p53 pathway is also frequently deregulated in GBM [98]. In primary and secondary GBM, TP53 mutation is observed in up to 30% and 70% of cases, respectively, which results in a common molecular abnormality linked to a worse prognosis [100]. Wang et al. [101] confirmed that *TP53* mutations are associated with poorer prognosis and shorter survival time in patients with GBM. In addition, *TP53* mutation may reduce the chemosensitivity of GBM to temozolomide by increasing *MGMT* expression [101].

Investigating the prevalence of *TP53* mutations in GBM, Homma et al. [102] identified *TP53* mutations in 113 out of 420 patients (26.9%). Interestingly, they observed a correlation between *TP53* mutations and a specific GBM subtype—giant-cell glioblastoma, characterized by atypical large cells with multilobed nuclei. Notably, *TP53* mutations were detected in 78% of giant-cell GBM cases. Conversely, TP53 mutations were not found in necrotic GBM [102].

Their analysis also revealed an association between *TP53* mutations and patient characteristics. Patients with *TP53* mutations tended to be younger and have secondary GBM (arising from pre-existing lower-grade gliomas) compared to those without the mutation proven [102].

Furthermore, the study explored the potential prognostic value of *TP53* mutation status. They divided patients into two groups based on survival: long-term survivors (over 3 years after surgery) and short-term survivors (under 1.5 years after surgery). Patients with higher p53 expression, a protein encoded by the *TP53* gene, were found in the long-term survival group (85%), compared to 56% in the short-term survival group [102]. These findings suggest a potential link between p53 expression and improved survival in GBM patients, warranting further investigation.

Cantero et al. [103] analyzed the genetic profiles of 36 glioblastoma patients. p53 expression was detected in all samples. *BRAF* and *H3F3A* mutations were uncommon or not detected at all. A set of 36 GBM samples was divided into two groups: wild-type GBM (wt-GBM) and giant-cell GBM (gc-GBM). In the giant-cell GBM group, the frequencies of p53 expression and *ATRX*, *RB1,* and *NF1* mutations were higher, while *EGFR* amplification, *CDKN2A* deletion, and *TERT* mutations were less frequent. Patients with gc-GBM with proven *TP53* mutation were found to have better survival rates than patients with wt-GBM and *TP53* mutation. gc-GBM has different molecular properties than wt-GBM, in addition to unusually common *ATRX* mutations, *EGFR* amplifications, and *CDKN2A* deletions [103].

Out of a group of 301 patients with GBM, Weller et al. [104] reported *TP53* mutation in 15%; the overall outcome was not affected by the presence/absence of *TP53* mutation [104].

## 4. Discussion

*ATRX*: From the studies examined above, we find that the overexpression of *ATRX* in glioma cells does not significantly affect patient prognosis or overall survival. In addition, *ATRX* mutations correlate with other markers. While *TERT* promoter mutation positively correlates with patient age and the co-present *IDH* mutation, it worsens overall survival in combination with *ATRX* mutation. On the other hand, the loss of ATRX expression correlates with better overall outcome. In a study by Cai et al. [25], the longest survival was in the group with loss of ATRX expression and confirmed *IDH1* mutation, a total of 19.6 months [25].

*BRAF*: *BRAF* mutations are more common in young people, in whom they are associated with better survival (Table 5).

*EGFR*: Studies confirm that *EGFR* mutations have a negative impact on the prognosis of patients with GBM (Table 5).

*IDH1*: *IDH1* mutation correlates with better overall outcome, and high-grade gliomas with proven IDH1 mutation are now reclassified according to the new WHO classification as astrocytomas grade 4 (Table 5).

*MGMT*: *MGMT* methylation is observed in 30–60% of GBM [105] and is often associated with a coexisting *IDH1* mutation. Patients with proven *MGMT* methylation and *IDH1* mutation benefit significantly from temozolomide treatment. As Table 5 shows, the average survival time of these patients can be over 35 months (Table 5).

*PIK3CA*, *PIK3R1*: *PIK3CA* mutations promote differential gliomagenesis depending on the mutated domain. Despite the association with younger age, PIK3CA-activating mutations are associated with earlier recurrence and shorter survival in adult patients with GBM. In addition, *PIK3CA*-mutant GBM has a greater tendency to disseminate [78]. The average overall survival time is around 20 months.

*PTEN*: *PTEN* is a tumor suppressor gene commonly inactivated in GBM, but the prognostic significance of the *PTEN* mutation remains controversial (see Table 5). PTEN protein overexpression is associated with shorter PFS and OS in patients with GBM [83].

*TERT*: The results of studies evaluating *TERT* mutation in GBMs as a negative prognostic marker (Table 5).

*TP53*: *TP53* mutations are associated with poorer prognosis and shorter survival times in patients with GBM. In addition, according to Wang et al. [65], *TP53* mutation reduces the chemosensitivity of tumor cells to temozolomide by increasing MGMT expression (Table 5).

We averaged the resulting values of overall survival in months from the table above (see Table 5) and divided patients into four groups according to survival: group A with an average survival of over 25 months; group B, 20–25 months; group C, 15–20 months; and group D, under 15 months (Table 6).

Table 6 shows that group A (best OS over 25 months) includes a group with proven *IDH1* mutation (now astrocytoma IDH mutant gr. 4); in group B (OS 20–25 months), we can include GBM with mutations in the *MGMT*, *BRAF*, *TP53*, *PIK3CA*, and *PIK3R1* genes. Group C (OS 15–20 months) includes loss of *ATRX* expression and *PTEN* mutations, and group D (OS less than 15 months) includes *TERT* mutation and *EGFR* amplification.

## 5. Conclusions

According to the latest WHO classification, a high-grade glioma can be classified as grade 4 glioblastoma when histologic features of malignancy such as necrosis and microvascular proliferation are present or when *TERT* promoter mutation, *EGFR* gene amplification, or +7/−10 chromosome copy number changes are detected.

Previous research has focused on studying isolated gene families within individual signaling pathways in glioblastoma patients. However, this approach may be insufficient for selecting the most appropriate therapeutic target. A more comprehensive approach utilizing massive parallel sequencing is needed to adequately identify potential therapeutic targets. Bioinformatics tools can then be leveraged to analyze these targets in the context of the patient’s immune microenvironment. PDL1 expression alone is an insufficient indicator of response to immunotherapy, highlighting the need for a more comprehensive evaluation.

Predicting a patient’s prognosis based on a single genetic alteration remains challenging. The current literature has identified mostly negative prognostic markers in the pathophysiology of glioblastoma. Therefore, the goal of my ongoing study is to focus on the long-term survival of glioblastoma patients, exploring the potential existence of genes that could serve as positive prognostic indicators, potentially improving patient outcomes.

We would like to create a panel of genes that could help us stratify patients with glioblastoma into prognostic subgroups, facilitating more personalized treatment approaches and improved patient outcomes.

## Figures and Tables

**Table 1 ijms-25-04438-t001:** Types of PTEN mutations and their impact on prognosis [83].

Type of *PTEN* Mutation		PFS (Months)
missense	51.20%	no effect
nonsense	16.90%	3.8
frameshift	24.90%	no effect
other (overexpression)	7%	7.2

**Table 2 ijms-25-04438-t002:** Differences in patient survival rates by age, Karnofsky score, and PTEN mutations (adjusted according to Carico et al., 2012) [84].

	Median of Overall Survival (Months)	Number of Patients	Percentage
Age (years)			
18–49	41	21	16.8
50–64	19.5	53	34.2
65	15	76	49
Karnofsky Performance Score			
80–100	25.7	108	69.7
50–70	9.8	42	27.1
under 40	6.3	5	3.2
*PTEN* mutation			
deletion	18.2	83	53.6
mutation	20	72	46.5

**Table 3 ijms-25-04438-t003:** Occurrence of C228T, C250T, and C229A mutations [92].

*TERT* mutation 92 (73%)	66 (72%)	C228T mutation
	26 (28%)	C250T mutation
	0 (0%)	C229A mutation

**Table 4 ijms-25-04438-t004:** Frequency of rs2853669 polymorphism occurrence [92].

rs2853669 polymorphism	59 patients (47%) “noncarriers”	
	67 patients (53%) “carriers”	12 homozygous CC
		55 heterozygous CT

**Table 5 ijms-25-04438-t005:** Mean survival of patients with proven mutation of selected genes.

	Author	Number of Patients	OS (Months)
*BRAF*	Zheng et al., 2021 [30]	16	16.8–27.8
	Wang et al., 2019 [34]	8	6.0
	Vuong et al., 2017 [31]	1308	9.8–28.1
	Takahaschi et al., 2015 [9]	1	48.0
	Chi et al., 2013 [32]	5	16.0–36.0
	Da et al., 2021 [33]	69	3.9–16.0
*EGFR*	Navarro et al., 2020 [41]	137	5.6–9.6
	Armocida et al., 2020 [42]	146	10.0–18.0
	Munoz-Hidalgo et al., 2020 [43]	46	12.0–14.0
*IDH1*	Polivka et al., 2018 [54]	30	4.6–13.3
	Goryanov et al., 2017 [57]	14	44.1
	Paldor et al., 2016 [58]	42	23.6–25.2
	Chen et al., 2016 [55]	570	30
*MGMT*	Myung et al., 2016 [62]	34	6.7–18.5
	Boots-Sprenger et al., 2013 [63]	15	13.0–15.0
	Kamoshima et al., 2012 [64]	5	36
	Wang et al., 2014 [65]	78	15.6
	Yang et al., 2015 [66]	274	35.8
*PTEN*	Koshiyama et al., 2017 [82]	40	4.8
	Xu et al., 2014 [83]	586	3.8–7.2
	Carico et al., 2012 [84]	155	18.2–20.0
	Ruano et al., 2009 [88]	194	10.0
	Ermoian et al., 2002 [87]	46	45.5
	Backlund et al., 2003 [89]	129	5.5
*TERT*	Mosrati et al., 2015 [93]	92	11.0–20.0
	Kikuchi et al., 2020 [95]	147	7.0–10.0
	Simon et al., 2015 [94]	143	15.0–25.0
	Nonoguchi et al., 2013 [91]	358	9.3–10.5
*TP53*	Wang et al., 2014 [65]	68	8.2
	Homma et al., 2006 [102]	420	7.9–14.2
	Weller et al., 2009 [104]	301	6.8–12.5
	Cantero et al., 2020 [103]	36	12.0

**Table 6 ijms-25-04438-t006:** Mean survival and mutation frequency in glioblastoma.

Gene Mutation	OS (Months)	Frequency [105]
*ATRX* (loss of expression)	19.6	6%
*BRAF*	21.8	1.7%, 50% epitheloid type
*EGFR* (amplification)	11.5	40–50%
*MGMT*	22.8	30–60%
*PIK3CA, PIK3R1*	20.0	9–13%
*PTEN*	15.1	41%
*TERT*	13.5	50–74%
*TP53*	20.5	28%

## Data Availability

All the date used in this manuscript are available on Pubmed.

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
