# Peer review of "Potential Diagnostic and Clinical Significance of Selected Genetic Alterations in Glioblastoma"

_ijms, 2024, doi:10.3390/ijms25084438_

Round 1

Reviewer 1 Report

Comments and Suggestions for Authors

In the manuscript, the authors suggested the classification of genetic predictors of survival rates in glioblastoma patients. The authors chose to describe 10 gene candidates, where mutations were correlated to patient outcomes. They give a critical opinion on glioblastoma classification shortcomings and emphasize the importance of including more predictive markers in diagnosis and treatment.

Strong points: The concept of classifying genetic alterations based on outcome predictions of the disease in a way authors proposed appears original. 

Downsides: There are many attempts at defining the molecular and genetic predictors that could provide more context to disease and possible treatment strategies; the genes described here are not exactly a novelty. One of the obstacles in defining reliable predictors in glioblastoma is that the tumor is rare and variability among patients and tumor heterogeneity are quite represented, which authors acknowledged in the manuscript - please comment on how this (or any other similar) system could be implemented in treating patients. Importantly, the review is not reader-friendly. It needs to be organized better for the reader to be familiar with the main focus and purpose of the manuscript. Also, please add why you chose the specific 10 genes here described since more candidates have been proposed and are under research. 

Specific comments are as follows:

Please explain in the text why you find the here described genetic alterations as secondary.

References are missing for some claims mentioned as facts – eg. what is the source for the 14-month survival rate after the diagnosis?

At times, the paragraphs in the text are not very well connected. Eg, the authors write about vemurafenib; they continue to something else; later in the text, there is another mention of the same drug in the same context.

An integrated single table with highlighted important information, uniformed for all genes you chose to describe would significantly improve the manuscript's quality. Suggestions for classification are: is the genetic change loss or gain in function, is the gene mutation, a predictor of a positive or negative outcome (+ or -), the total patient number from the papers included per gene/if possible with details on age and gender, the overall survival, the classification letter, from your proposed system, references to research, etc. The tables at the current state are not mentioned in the text and are not very helpful for following the paper, since the information is fragmented and scattered. 

I suggest adding a graphic representation of manuscript-relevant genes - the interactions and/or overlapping signaling pathways.

I also suggest including a separate segment on how these gene alterations can be used in creating novel therapies, with referenced relevant clinical research, if there are any.

 “In many studies, ATRX gene alterations have been shown to be associated with prognosis and mostly correlate with favorable results [6]” – you say “many studies” but then refer to one research. 

Comments on the Quality of English Language

The English language in the manuscript in general is clear and comprehensible. With minor writing style corrections, it could be improved.

Author Response

Good evening,

Thank you for reading my article and for the valuable advice that helped to improve the quality of the upcoming publication. I tried to edit the text according to the instructions - supplemented or edited parts are in blue in the text.

What was improved according to the recommendations:

  • genetic alterations as secondary
  • By secondary alterations we mean mutations induced by cancer treatments such as radiotherapy or chemotherapy.
  • references are missing for some claims mentioned as facts – eg. what is the source for the 14-month survival rate after the diagnosis?
  • The reference was completed (see text).
  • paragraphs in the text are not very well connected. Eg, the authors write about vemurafenib; they continue to something else; later in the text, there is another mention of the same drug in the same context.
  • We tried to ensure a cohesive flow of ideas throughout the text.
  • integrated single table with highlighted important information, uniformed for all genes you chose to describe would significantly improve the manuscript's quality.
  • See Table 5 with integrated details.
  • graphic representation of manuscript-relevant genes - the interactions and/or overlapping signaling pathways.
  • The manuscript has been expanded to include a section on signaling pathways.
  • separate segment on how these gene alterations can be used in creating novel therapies, with referenced relevant clinical research, if there are any.
  • Predicting a patient's prognosis based on a single genetic alteration remains challenging and therefore in the future we would like to create a panel of genes that could help us stratify patients with glioblastoma into prognostic subgroups, facilitating more personalized treatment approaches and improved patient outcomes. This is the aim of my research and I hope very soon will some results presented on our homogenous patient cohort of 100 GBM patients who all underwent total resection surgery followed by standardized radiation and chemotherapy.

Reviewer 2 Report

Comments and Suggestions for Authors

Title:

Potential Diagnostic and Clinical Significance of Selected Genetic Alterations in Glioblastoma Multiforme

 Summary:

Survival outcomes in glioblastoma multiforme are intricately tied to its molecular biological properties, drawing interest from the medical community. The authors systematically reviewed a series of selected studies published between 2016 and 2022, focusing on particular genes. The conclusion posits that mutations within specific gene groups exert divergent effects on prognosis and quality of survival in glioblastoma patients.

 Major Comments:

The review's utility diminishes without reference to the latest WHO classification. Outdated references to primary or secondary glioblastoma or glioblastoma grade IV render survival data obsolete.

While the authors compile a review of existing articles grouped by relevant mutations, the presentation lacks clarity, featuring a mix of works varying in quality. The text intertwines diverse aspects, leading to confusion. Certain statements are unclear, with factual inaccuracies. For instance, attributing a better prognosis to GBMs with a confirmed IDH1 mutation and loss of ATRX expression fails to distinguish mutation from EGFR amplification. Additionally, the authors incorrectly conflate mutation and methylation of the MGMT gene promoter.

The conclusions lack clarity and fail to add value. Furthermore, generalizations about mutations cannot be applied to individual patients, as indicated by the authors, as they are applicable to cohorts rather than specific patients.

 Minor Comments:

1.Tables should be omitted from the conclusions.

2.Manuscript lines should be sequentially numbered.

3. The term "glioblastoma multiforme" is outdated; the neoplasm should be referred to as glioblastoma.

4.The review encompasses genetic and epigenetic alterations, such as MGMT.

5.The lack of adequate text revision is exemplified, for instance, by the incorrect reference to angioimmunoblastic carcinomas of T-cell lymphomas.

 Conclusion:

The authors' aim to provide an update on the molecular alterations of glioblastoma is appreciated. While the article offers a comprehensive evaluation of significant molecular alterations in glioblastoma, its quality falls short for publication. Numerous errors and inaccuracies detract from the writing's quality, and the study's objective lacks adequate argumentation. Therefore, the current article does not meet the standards for publication in the Journal.

Comments on the Quality of English Language

The quality of the writing is poor and should be improved.

The quality of the writing is poor and should be improved.

The quality of the writing is poor and should be improved.

The quality of the writing is poor and should be improved.

The quality of the writing is poor and should be improved.

The quality of the writing is poor and should be improved.

Author Response

Good evening,

Thank you for reading my article and for the valuable advices that helped to improve the quality of the upcoming publication. I tried to edit the text according to the instructions - supplemented or edited parts are in blue in the text.

What changes were made according to the recommendations:

Our manuscript has undergone revisions to reflect the latest terminology and enhance reader comprehension. Notably, outdated terms like "glioblastoma multiforme" have been replaced throughout the text. Additionally, we have incorporated relevant details regarding the new WHO classification system.

For improved clarity and organization, tables have been relocated from the conclusion section. The conclusion itself has been expanded to include a new section exploring potential signaling pathway interactions.

Furthermore, we have adopted a more reader-friendly approach in certain sections, aiming to improve accessibility for a broader audience.

This revision aligns with the ongoing research in our group, which focuses on identifying interactions between selected genes in GBM. Our ultimate goal is to develop a gene panel that facilitates the stratification of patients into prognostic subgroups. This approach holds promise for personalized treatment strategies and improved patient outcomes.

Round 2

Reviewer 1 Report

Comments and Suggestions for Authors

I confirm the quality of work is majorly improved after revision.

Major concerns remain:

I would like to draw attention to the iTheticate result: The TP53 section is copied/pasted from https://www.wikilectures.eu/w/P53 – the copy/pasted section has to be re-written by the authors. Copy/pasted sentences from other published works are also found in this manuscript - the work they are copied from needs to be appropriately referenced, and the sentences under quotation marks.

Please confirm the Abstract is not AI-generated. Also, the references are not necessary in the Abstract; my previous comment was on the account of the Introduction. Reference numbers should be in order of appearance.

 “These genes are involved in various cellular processes. “ – the sentence is vague and undetermined.

The text on the signaling pathways is more appropriate as an addition to the rationale as to why you chose the specific genes here described, and it is more appropriate in the Introduction, not the conclusion. However, I do agree the previous conclusion should be expanded – by expanding your critical opinion, supported by the text in the manuscript, on the subject you chose to review.

Author Response

Good afternoon,

thank's for the previous comments on the manuscript. It's great to hear the quality of it improved after revision.

The part about TP53 mutation was completely changed.

Yes, I confirm, the abstract was not AI generated. It was written in czech language and translated to english by professional translating company. 

The sentence  “These genes are involved in various cellular processes. “ was succesfully removed. 

And like suggested, the part about signalling pathways is now in the introduction section.

I have written my opinion in the conclusion - what we actually woult like to prove with our ongoing research - I focus on long-term survivors. All my patients in the study (100 of them) were operated from 2016-2022, all of them have undergone total surgical resection of primary IDH wild type glioblastoma, all received adjuvant radiation and chemotherapy. We would like to create a panel of genes that could help us stratify the patients into prognostic subgroups, predicting survival and a response to treatment.

And the last but not least, references are in ascending order now.  

I greatly appreciate your time and comments while reading this article, I take them as constructive criticism that can help me move towards my desired goal :-)

With kind regards

Silvia Tomoszková

Round 3

Reviewer 1 Report

Comments and Suggestions for Authors

The manuscript can be published in its present form, with a minor correction:

In the conclusion: " Therefore,  the  goal  of  our  ongoing  study  is  to  focus  on  long-term  survival glioblastoma patients,..."

Thank you for acknowledging the comments. They are meant as constructive criticism. All the best with the research :)